

# Asymmetric responses to simulated global warming by populations of *Colobanthus quitensis* along a latitudinal gradient

Ian S. Acuña-Rodríguez[1], Cristian Torres-Díaz[2], Rasme Hereme[1] and Marco A. Molina-Montenegro[1,3,4]

[1] Centro de Ecología Molecular y Aplicaciones Evolutivas en Agroecosistemas (CEM), Instituto de Ciencias Biológicas, Universidad de Talca, Talca, Chile
[2] Laboratorio de Genómica y Biodiversidad (LGB), Departamento de Ciencias Básicas, Universidad del Bío-Bío, Chillan, Chile
[3] Centro de Estudios Avanzados en Zonas Áridas (CEAZA), Facultad de Ciencias del Mar, Universidad Católica del Norte, Coquimbo, Chile
[4] Research Program "Adaptation of Agriculture to Climate Change" PIEI A2C2, Universidad de Talca, Talca, Chile

Corresponding author
Marco A. Molina-Montenegro, marco.molina@utalca.cl

## ABSTRACT

The increase in temperature as consequence of the recent global warming has been reported to generate new ice-free areas in the Antarctic continent, facilitating the colonization and spread of plant populations. Consequently, Antarctic vascular plants have been observed extending their southern distribution. But as the environmental conditions toward southern localities become progressively more departed from the species' physiological optimum, the ecophysiological responses and survival to the expected global warming could be reduced. However, if processes of local adaptation are the main cause of the observed southern expansion, those populations could appear constrained to respond positively to the expected global warming. Using individuals from the southern tip of South America, the South Shetland Islands and the Antarctic Peninsula, we assess with a long term experiment (three years) under controlled conditions if the responsiveness of Colobanthus quitensis populations to the expected global warming, is related with their different foliar traits and photoprotective mechanisms along the latitudinal gradient. In addition, we tested if the release of the stress condition by the global warming in these cold environments increases the ecophysiological performance. For this, we describe the latitudinal pattern of net photosynthetic capacity, biomass accumulation, and number of flowers under current and future temperatures respective to each site of origin after three growing seasons. Overall, was found a clinal trend was found in the foliar traits and photoprotective mechanisms in the evaluated *C. quitensis* populations. On the other hand, an asymmetric response to warming was observed for southern populations in all ecophysiological traits evaluated, suggesting that low temperature is limiting the performance of *C. quitensis* populations. Our results suggest that under a global warming scenario, plant populations that inhabiting cold zones at high latitudes could increase in their ecophysiological performance, enhancing the size of populations or their spread.

## INTRODUCTION

The Antarctic continent is among the most stressful environments for plant life worldwide (*Robinson, Wasley & Tobin, 2003*; *Peck, Convey & Barnes, 2006*); establishment and survival is limited by conditions such as low temperatures, desiccation, wind abrasion, high radiation and low water and nutrient availability (*Alberdi et al., 2002*; *Robinson, Wasley & Tobin, 2003*; *Wasley et al., 2006*; *Convey, 2011*). Although it was recently indicated that warming has stopped in Antarctica (*Turner et al., 2016*), over the last few 50 years the mean annual temperature of the Antarctic Peninsula increased by almost 3 °C (*Vaughan et al., 2003*; *Turner et al., 2014*). Although global warming is a major threat for biodiversity worldwide, the Antarctic region is particularly sensitive to small increases in temperature, showing changes long before they can be seen elsewhere in the world (*Walther et al., 2002*; *IPCC, 2014*).

Only two vascular plants (*Deschampsia antarctica* and *Colobanthus quitensis*) have been able to establish and survive in Antarctica. Although both species are present in the same spatial range in Antarctica (from 62°S to 68°S), *C. quitensis* is more restricted in its habitat distribution than *D. antarctica*. *C. quitensis* (Kunth) Barttl. (Caryophyllaceae), commonly known as the Antarctic pearlwort, is a small-sized cushion-like perennial herb, with self-compatible sexual reproduction (*Kennedy, 1993*; *Convey, 1996*). The Antarctic pearlwort has an extremely wide range of distribution spanning from Mexico (17°N) to the southern Antarctic Peninsula (69°S) (*Smith, 2003*; *Convey, 2012*). Recent warming during the last five decades has produced new summer ice-free areas, which have provided suitable habitats for plant colonization (*Convey et al., 2014*; *Cannone et al., 2016*). Consequently, increases in both the size and number of *C. quitensis* and *D. antarctica* populations have been reported (e.g., *Smith, 1994*; *Day et al., 1999*; *Torres-Mellado, Jaña & Casanova-Katny, 2011*; *Cannone et al., 2016*), and southward population expansions can be projected for the next century. Although climate change is expected to have an overall positive impact on the growth, survival and fitness of *C. quitensis* (*Convey, 2011*; *Day et al., 2009*; *Molina-Montenegro et al., 2012a*; *Torres-Díaz et al., 2016*), the eco-physiological responses to the different components of climate change (e.g., warming) could differ among populations along the latitudinal gradient.

In species with widespread distribution ranges, peripheral populations such as those of *C. quitensis* from the Antarctic Peninsula (ca. 68°S) are expected to depart more from the species' physiological optimum than their central or northern counterparts (*Gaston, 2009*; *Sexton, Hangartner & Hoffman, 2014*). This prediction finds support in previous studies showing that *C. quitensis* is limited by abiotic conditions (e.g., low temperatures); this limitation is more evident in the southern populations of its range (*Gianoli et al., 2004*; *Sierra-Almeida et al., 2007*; *Torres-Díaz et al., 2016*). For instance, *Sierra-Almeida et al. (2007)* found significantly higher net photosynthesis rates in Antarctic (62°S, King George Islands) than in Andean (33°S, La Parva) populations of *C. quitensis* at both low
and high temperatures (4 and 15 °C), suggesting that Antarctic populations of *C. quitensis* are physiologically adapted to colder habitats and that thermal stress release would produce increases in their performance. With respect to cold adaptation, *Gianoli et al. (2004)* found greater levels of freezing resistance after cold acclimation and shorter and wider leaves in Antarctic than Andean genotypes. The conjunction of high irradiance and low temperature may damage the photosynthetic apparatus, causing a reduction in photosynthesis known as photoinhibition (*Demmig-Adams & Adams, 1992*). *Bascuñan-Godoy et al. (2010)* showed that foliar microstructures as well as high levels of the xanthophyll cycle pool helped to maintain high physiological performance in *C. quitensis* from an Antarctic population more than an Andean population under high radiation and low temperature conditions. Thus, differential responses in morphology and/or physiology can be expected among populations of *C. quitensis* along the latitudinal distribution gradient, based on the most recent global warming predictions and models (*IPCC, 2014*; *Turner et al., 2014*).

Since Antarctic fieldwork can be complex and logistically demanding, most of the experimental studies of global warming have been carried out under controlled laboratory conditions using present and future climate scenarios (*Molina-Montenegro et al., 2016*, but see: *Day et al., 2008* and *Day et al., 2009*). Most studies assessing the effects of warming on different plant species along a latitudinal gradient use a constant rise in temperature for all origins, although warming is specific to every place (*IPCC, 2014*). Thus, it is common to observe several studies assessing the effects of global warming on Antarctic plant species that have used only one temperature to represent the current conditions of the Antarctic ecosystem—Peninsula or Maritime Antarctica—as well as for the projected warming. But since a strong abiotic stress increase is present along the latitudinal gradient from the South Shetland Islands to the Antarctic Peninsula, it generates an important bias for the local conditions compared to the regional averages (*Vaughan et al., 2003*). For this reason, in order to make a more realistic analysis of the responses of Antarctic vascular plants to global warming, experimental studies should be performed considering their current and projected site-specific temperatures.

The main goals of this study were to determine whether populations of *C. quitensis* possess latitudinal variation in several traits (foliar anatomy and pigments) that avoids photoinhibition and to determine whether the ecophysiological response to simulated climate change will differ among populations distributed along a latitudinal gradient. We specifically addressed the following questions: (1) Does *C. quitensis* show signs of clinal variation in anatomical (foliar microstructures) or physiological (xanthophyll pigments) traits along a latitudinal gradient in which environmental stress (the combination of cold, aridity and photo-inhibitory radiation) increases with latitude; and (2) Will southern populations of *C. quitensis* be more responsive to simulated global warming than more northern populations? To address these questions, we measured leaf anatomy and xanthophyll pigments in *C. quitensis* along a latitudinal gradient consisting of three locations from South America to the Antarctic Peninsula. In addition, we measured the change of the photosynthetic responses, total biomass and flower production in *C. quitensis* from all locations exposed for three years to simulated global warming.

## MATERIALS AND METHODS

### Target species and study sites

Commonly known as the Antarctic pearlwort, *Colobanthus quitensis* (Kunth) Bartl. (Caryophyllaceae), is a small-sized cushion-like perennial herb with an extremely wide distribution range spanning from Mexico (17°N) to the Antarctic continent (68°S); sporadic populations can be found in different islands of Maritime Antarctica, as well as along the coast of the Antarctic Peninsula (*Smith, 2003*). *C. quitensis* individuals in Antarctic ecosystems are mostly distributed near seashores which are frequently associated with *D. antarctica* and mosses. Clonal reproduction is the more common means of propagation in the Antarctic populations of *C. quitensis*, but it is also capable of self-compatible sexual reproduction (*Convey, 1996*).

Individuals of *C. quitensis* were collected from three locations along a simple latitudinal transect: South America (SA), close to the city of Punta Arenas (53.1°S–70.9°W); South Shetland Islands (SI), close to the Polish Antarctic Station in Admiralty Bay (62.1°S–58.3°W), and the Antarctic Peninsula (AP), in Lagotellerie Island (67.5°S–67.2°W) (Fig. 1). Sixty plants were dug out from each site during the 2014/2015 growing season with enough soil around the roots (ca. 250 g) and kept well-watered in a plastic box under natural conditions of light and temperature until their transportation to the growth chambers at the Universidad de Talca, Chile (35.2°S). All plants were collected under permission of the Chilean Antarctic Institute (INACH; authorization number: 1060/2014).

### Latitudinal trait variation in pigments and microstructures

Variations in anatomical (foliar microstructure) and physiological (photoprotective pigment) traits were assessed in individuals from all locations along the latitudinal gradient. Specifically, in 25 individuals from each location (one leaf per plant) we measured and compared five microstructure foliar traits: cuticle, mesophyll, palisade parenchyma and spongy mesophyll width (mm), and leaf transversal area (mm$^2$). In the laboratory, leaves were sectioned in 2–3 mm in length and dehydrated in a graded ethanol series. Thin fragments of 75–100 nm sections were prepared on an ultra-microtome. Sections were stained with toluidine blue and viewed with a light microscope to analyse morphological attributes of the leaf. Thin sections were cut with a diamond knife and were stained on grids with uranyl acetate and lead citrate, and then examined in transmission electron microscope.

In addition, in a subset of 15 individuals from each location (one leaf per plant), we estimated the xanthophyll-cycle pigment content (violaxanthin, antheraxanthin and zeaxanthin) and the de-epoxidation state of the xanthophyll cycle pigments (DEPS). Leaves were repeatedly extracted (three times) with ice-cold 85% (v/v) acetone and 100% acetone using sonication for 45 min at 4 °C. Pigments were separated on a Dupont non-end capped Zorbax ODS-5 μmol column at 30 °C at a flow rate of 1 ml min-1. The solvents consisted of (A) acetonitrile/methanol (85:15, v/v) and (B) methanol/ethyl acetate (68:32, v/v). The HPLC gradient used was: 0–14 min 100% A, 14–16 min decreasing to 0% A, 16–28 min 0% A, 28–30 min increasing to 100% A, and 30–38 min 100% A. Detection was carried out at 445 nm. The de-epoxidation state (DEPS) of the pigments involved in the xanthophyll

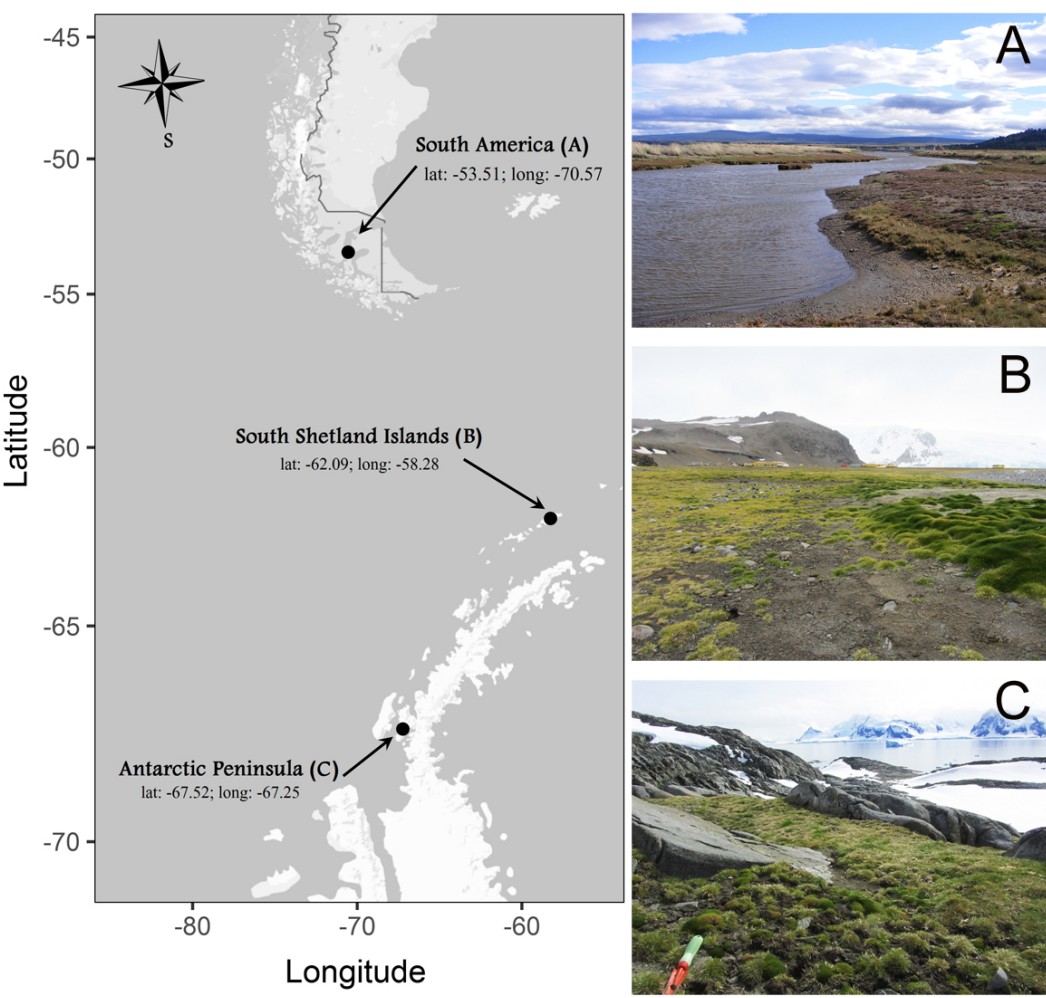

**Figure 1** **Sites where individuals of *Colobanthus quitensis* were collected: (A) South America (SA: 53.5°S), (B) Shetland Islands (SI: 62.1°S) and (C) Antarctic Peninsula (AP: 67.5°S).** Locations photographed by Marco A. Molina-Montenegro on 2012–2013 growing season.

cycle was quantified as $Z + (0.5 \times A)/(V + A + Z)$ (for more details see *Molina-Montenegro et al., 2012b*).

## Latitudinal responses to simulated global warming experiment

To evaluate whether different latitudinal origins of *C. quitensis* differ in the magnitude of their responses to global warming, we measured the ecophysiological performance of plants from each origin (South America, Shetland Islands and Antarctic Peninsula) under current and future temperature conditions (warming) predicted by climate change models (*IPCC, 2014*). Based on the available predictions, a 4 °C increase in temperature was applied as warming treatment for all populations. Thus, "current" temperatures were 10 °C, 5 °C and 3 °C for SA, SI and AP, and "warming" temperatures were 14 °C, 9 °C and 7 °C for SA, SI and AP, respectively.

We established six experimental abiotic conditions using six automatic air-cooling growth chambers (model: LTJ300LY; Tianyi Cool, China) to simulate current and

future environmental conditions during summer months of each latitudinal origin. The current conditions for each location were: South America (mean temperature: 10 °C, photoperiod: 18/6 h light/dark), Shetland Islands (mean temperature: 5 °C, photoperiod: 19/5 h light/dark) and Antarctic Peninsula (mean temperature: 3 °C, photoperiod: 21/3 h light/dark). In addition, future conditions for each location were: South America (mean temperature: 14 °C, photoperiod: 18/6 h light/dark), Shetland Islands (mean temperature: 9 °C, photoperiod: 19/5 h light/dark) and Antarctic Peninsula (mean temperature: 7 °C, photoperiod: 21/3 h light/dark). During all experimental time every chamber was maintained in the temperature and photoperiod indicated above, with a constant intensity of the photosynthetic active radiation (PAR) of 275 $\mu$mol m$^{-2}$s$^{-1}$. But PAR was lowered to 20 $\mu$mol m$^{-2}$s$^{-1}$ during seven months in order to mimic the natural variation in the solar radiation that affect at high latitude populations. A total of 30 plants per each latitudinal origin (SA, SI and AP) were randomly assigned to the current or warming conditions. Hence, in each growth chamber a total of 15 plants were maintained during three years, and once per year in January (middle of the growing season), photosynthetic rate was recorded on every individual in both current and warming conditions. The photosynthetic rate was recorded using an infra-red gas analyzer (IRGA, model Ciras-2; PP-System, Amesbury, MA, USA) under the same temperature of each growth chamber. In addition, at the end of the third year two fitness-related traits were measured as response variables, total biomass accumulation and reproductive effort. Total plant growth was measured as the average individual total biomass increase (final biomass—initial biomass, g) during the three years of the experiment. At the beginning of the experiments, we selected similar size plants (from each origin). These plants were randomly assigned to current and warming treatments. There were no differences in the initial weight of plants between current and warming treatments for any latitude (one-way ANOVA, $F_{2,87} = 29, 4$; $p < 0.001$). At the end of the experiments, whole plants were harvested (root plus shoot), and individually weighed using a digital balance. Reproductive effort was measured by counting the number of flowers produced by each plant during the last year of the experiment. Since there were no differences in initial plant biomass (fresh weight) between current and future thermal regimes, all individuals were weighed before the start of the experiment. The magnitude of the responses of each latitudinal origin of *C. quitensis* to experimental warming was estimated as the average difference between future and current thermal conditions (i.e., delta = future—current values).

## Statistical analysis

We used one-way ANOVAs to assess the significance of the observed differences between origins for all microstructural traits, pigment content and the DEPS state of foliar samples of *C. quitensis*. The magnitude of the responses of *C. quitensis* to experimental warming on the delta values (future—current conditions) for each trait (photosynthetic rate, total biomass accumulation, and reproductive effort) in different populations along of the latitudinal gradient was tested using one-way ANOVAs. In all analyses, significant differences between treatments were estimated using Tukey tests (HSD) as *post-hoc* comparisons. Due to the

**Table 1  Leaf microstructure characteristics and mean pigment concentrations in foliar tissues of *Colobanthus quitensis* from three origins along a sub-Antarctic—Antarctic latitudinal gradient.** Values are means ± 2 SE. Different lowercase letters indicate significant differences between populations (Tukey HSD tests, $\alpha = 0.05$). Letters were ordered from higher to lower trait values (from a to c).

| Trait | South America (53.5°S) | Shetland Islands (62.1°S) | Antarctic Peninsula (67.5°S) |
|---|---|---|---|
| *Microstructure* | | | |
| Cuticule width (μm) | 19.54 ± 2.48[a] | 16.16 ± 1.59[b] | 11.52 ± 1.68[c] |
| Mesophyll width (μm) | 358.28 ± 23.99[c] | 408.96 ± 7.62[b] | 454.01 ± 11.28[a] |
| Palisade Parenchy ma width (μm) | 91.72 ± 1.48[c] | 95.92 ± 1.52[b] | 107.56 ± 5.09[a] |
| Spongy Mesophyll width (μm) | 246.92 ± 24.68[c] | 296.88 ± 8.41[b] | 334.92 ± 12.52[a] |
| Leaf transversal area (mm²) | 0.43 ± 0.03[a] | 0.33 ± 0.02[b] | 0.25 ± 0.02[c] |
| *Pigment contents* | | | |
| Anteraxanthin (μg g⁻¹ DW) | 73.73 ± 6.98[ab] | 77.06 ± 2.21[a] | 71.73 ± 3.03[b] |
| Violaxanthin (μg g⁻¹ DW) | 16.86 ± 2.64[b] | 18.13 ± 2.13[ba] | 19.46 ± 1.18[a] |
| Zeaxanthin (μg g⁻¹ DW) | 12.00 ± 3.29[c] | 24.46 ± 6.64[b] | 32.33 ± 5.89[a] |
| DEPS | 0.48 ± 0.02[c] | 0.52 ± 0.27[b] | 0.55 ± 0.02[a] |

number of multiple comparisons the sequential Bonferroni correction was applied to all *a posteriori* contrasts. All statistical tests were made using R v.3.1.3 (*R Core Team, 2015*).

## RESULTS

### Latitudinal trait variation in pigments and microstructures

As predicted, pigment contents (xanthophyll-cycle pigments), de-epoxidation state of pigments (DEPS) and micro-structural foliar traits showed a clinal pattern of variation (Table 1 and Table S1). With the exception of antheraxanthin that did not have a clear relationship with latitude, concentration of the other pigments and the DEPS significantly increased with latitude (Table 1). Leaves became significantly thicker and more cylindrical towards higher latitudes (Table 1 and Table S1). This pattern of clinal variation is consistent with the increase in the photo-inhibitory conditions (lower temperatures and moisture and longer daylight exposition) at higher latitudes.

### Latitudinal responses to simulated global warming experiment

The photosynthetic rates of *C. quitensis* exposed to experimental warming were significantly different among latitudinal origins (Figs. 2A–2C; Tables S2, S3). The greatest response to experimental warming was found in individuals from the southernmost population (AP), while individuals from South America showed the lowest response to experimental warming (Fig. 2; Tables S2, S3).

At the end of third year, experimental warming also had a greater positive and significant effect on total biomass accumulation and reproductive effort of Antarctic Peninsula *C. quitensis* individuals than those from South America or even from the Shetland Islands (Fig. 3; Tables S2, S3), suggesting that the observed differences in the photosynthetic response are also effectively translated into important fitness-related traits.

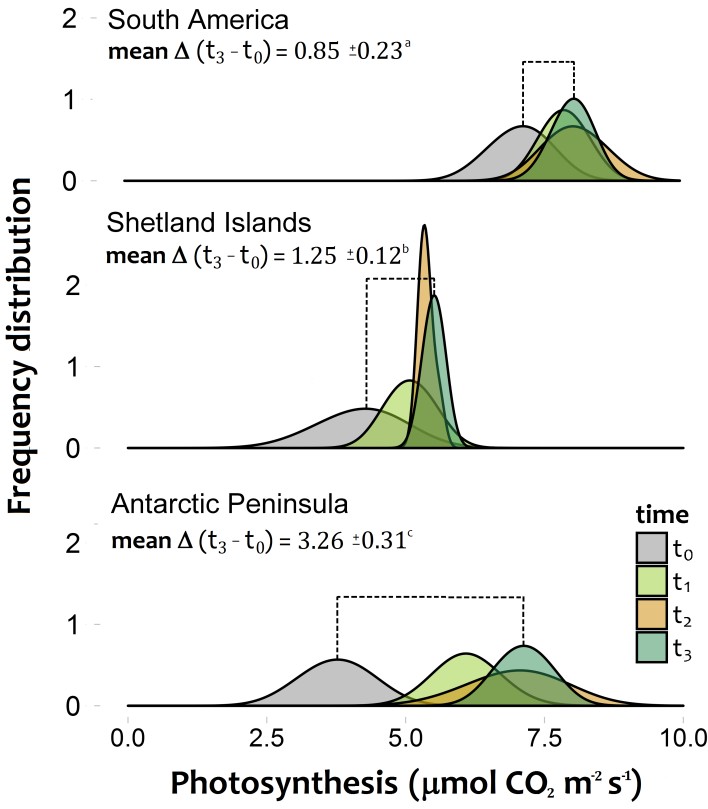

**Figure 2** **Frequency distribution of the photosynthetic response of *Colobanthus quitensis* individuals from three different origins (South America 53.5°S, Shetland Islands 62.1°S and Antarctic Peninsula 67.5°S).** Estimations were performed under both the current thermal conditions of each site ($t_0$), and during three consecutive simulated growing seasons ($t_1 : t_3$), where all individuals experienced their respective projected temperatures. The extent of the mean temporal response for each population is expressed as the mean delta value ($\pm$SD) between the photosynthetic responses at the last simulated warming season ($t_3$) and their values under the current thermal conditions ($t_0$). Different letters between these values refer to significant differences (Tukey HSD tests, $\alpha = 0.05$) between populations. Bonferroni correction was applied due to multiple comparisons.

# DISCUSSION

Our results indicate that in the widely distributed *C. quitensis*, morphological and physiological traits reflect an asymmetric response associated with increasing environmental stress induced by a combination of increasingly colder, arid and photo-inhibitory conditions with latitude. Moreover, photosynthetic performance and fitness-related traits were increased with warming in all populations. The prediction that responses to experimental warming would differ over latitude was supported by the greater responses in photosynthesis, biomass and reproductive effort found in the southernmost population of *C. quitensis*. Together, our findings suggest that the direction and magnitude of the responses of *C. quitensis* to global warming will be positive and specific to latitude.

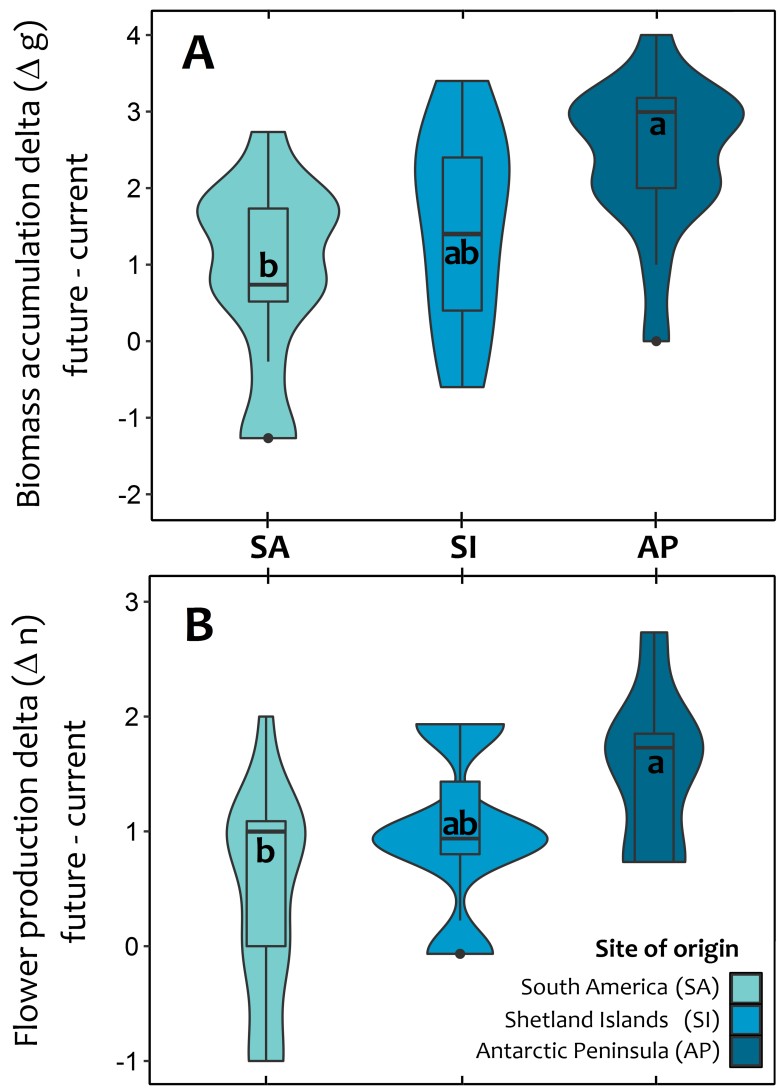

**Figure 3** **Violin plot for the average response of Colobanthus quitensis from three different latitudinal origins: South America (SA: 53.5°S), Shetland Islands (SI: 62.1°S) and Antarctic Peninsula (AP: 67.5°S) to experimental warming.** The effects of warming (+4 °C) were estimated as the absolute difference (i.e., delta) between plant performance under warming (simulated future conditions) and current thermal conditions in two fitness-related traits: aboveground biomass accumulation delta in grams (A) and reproductive effort delta estimated as the number of produced flowers (B). The mean delta value for each population in both traits was obtained after averaging the mean differences of each individual against all other plants from the same population. The box and whisker plot inside the violins represent the interquartile distribution of the data around the median (box inner line). Black dots correspond to outlier values. Different letters indicate significant differences (Tukey HSD tests, $a = 0.05$) between populations.

## Latitudinal trait variation in pigments and microstructures

The prediction of a progressive latitudinal change in both physiological and morphological traits was supported by seven of the eight traits measured in *C. quitensis*. The direction of the trait variation was consistent with the functional physiological responses expected to cope with increasingly stressful abiotic conditions such as cold temperatures, low

water availability and longer daylight periods found in the Antarctic continent (*Alberdi et al., 2002*; *Convey, 2006*). Despite the clear clinal variation in functional traits shown by *C. quitensis* that cope with photoinhibitory conditions, the biological performance (biomass and flower production) was lower in the southern margin. This result seems to agree with the core of the "spatial ecology theory" (*Gaston, 2009*; *Sexton, Hangartner & Hoffman, 2014*), which states that peripheral populations reduce their biological performance as a consequence of the reductions in habitat suitability and increased isolation (see, *Sagarin & Gaines, 2002*; *Vaupel & Matthies, 2012*).

Several studies have documented that high levels of the xanthophyll cycle pool could be considered as a pivotal mechanisms to avoid photoinhibition under high radiation and/or low temperature (*Demmig-Adams & Adams, 1996*; *García-Plazaola, Matsubara & Osmond, 2007*; *Molina-Montenegro et al., 2012b*). In addition, high levels of the xanthophyll cycle pool have been correlated with thermal dissipation of excess energy, being a dynamic and reversible mechanism in plants that inhabit cold environments (*Bravo et al., 2007*; *Bascuñan-Godoy et al., 2010*; *Molina-Montenegro et al., 2012b*; *Míguez et al., 2015*). In our study, those individuals of *C. quitensis* living in the south edge of distribution showed a significantly higher xanthophyll cycle pool, suggesting that in this population the capacity to avoid photoinhibition and to maintain its growth rate could be explained -at least in part- by the presence of this mechanism. The de-epoxidation state was also higher in the southern populations, where environmental stress is greater. The high de-epoxidation state found in all *C. quitensis* populations may be related to the capacity to dissipate excess energy, avoiding photoinhibition; this was more evident in the southern population where a combination of environmental stress conditions (colder, dryer, and photo-inhibitory) is found.

Morphological adaptations have been documented as one of the main mechanisms in plants to cope with environments (*Mooney et al., 1991*; and references therein). Foliar modifications in *C. quitensis* exposed to different abiotic conditions have been previously reported (*Bascuñan-Godoy et al., 2010*; *Cavieres et al., 2016*). We found that toward higher latitudes, *C. quitensis* individuals showed lower leaf transversal area and greater mesophyll thickness. It has been reported that increases in leaf thickness under cold acclimation may be beneficial for leaf survival under frost-induced cell dehydration, reducing the mechanical stress experienced during thawing (*Stefanowska et al., 1999*). The thicker palisade parenchyma shown by the southern population is likely to be associated with the higher photosynthesis values, since this foliar adaptation can influence $CO_2$ diffusion and assimilation (*Vieira & Mantovani, 1995*). These variations in foliar micro-structures seem to be a constitutive adaptation, since trait differences between populations were partially maintained in a common garden under controlled laboratory condition, suggesting that these functional adaptations could have a genetic basis.

## Latitudinal responses to global warming

Altitudinal and latitudinal shifts in distribution are among the earlier impacts of global warming observed in plants (e.g., *Menzel et al., 2006*; *Kelly & Goulden, 2008*; *Kopp & Cleland, 2014*). Nonetheless, there are still few studies focusing on the potential adaptive responses of plant species to climate change (but see *Nicotra et al., 2010*; *Merilä &*
*Hendry, 2014*). If increased temperature associated with climate change exceeds the thermal tolerance of a plant species, the species may either: (1) adjust to the new environmental conditions through acclimation (through phenotypic plasticity) or evolve (ecotype differentiation due to natural selection) (*Bellard et al., 2012*); or (2) track suitable environmental conditions in space (changing their geographical distribution) and/or time (adjusting their phenology and physiology). In contrast, if increased temperatures do not exceed the thermal tolerance, plastic responses rather than ecotype differentiation should be expected. Overall, our results indicate that *C. quitensis* will mainly respond to future warming through plastic adjustment and probably through range shifts after local scale population expansions in response to global change. However, biotic interactions (mostly competitive) will be also relevant to predict how this species will deal with future climate change. In a recent study, *Torres-Díaz et al. (2016)* showed that responses of *C. quitensis* to future climate change will be not only modulated by abiotic factors such as temperature and soil moisture, but also by biotic interactions with fungal endophytes. Moreover, plant–plant interactions between *C. quitensis* and the native and invasive plants could be even more critical to forecast the impacts of climate change on Antarctic vegetation. As shown by *Molina-Montenegro et al. (2016)* the alien *P. annua* shows high adaptive plasticity and greater competitive ability than the native *D. antarctica*. Therefore future studies should address how climate change will affect the interactions between the native species and also between the native *C. quitensis* and the alien *P. annua*.

Although environmental stress releasing due to future warming may promote anatomical changes that would increase traits such as light absorption, photosynthesis, growth and reproductive effort, warming may also compromise plant freezing tolerance and photoprotection. Warming has been reported to reduce the ability of perennial plants to resist freezing events through cold de-acclimation (reviewed in *Pagter & Arora, 2013*). Similar effects have been found in Andean plants exposed to experimental warming (e.g., *Sierra-Almeida & Cavieres, 2010*). Thus, future studies should address these potentially negative effects of stress releasing on Antarctic vascular plants.

Antarctic vascular plants are distributed along one of the most stressful gradients of the world, being affected by extreme environmental pressures that often limit their individual performance and population dynamics (*Peck, Convey & Barnes, 2006*). Nevertheless, recent warming during the last decades has already increased their frequency and distribution (*Cannone et al., 2016*). In a recent review, *Valladares et al. (2014)* showed how plastic responses are highly variable between populations of the same species, and how the spatial distribution of these responses are key to cope with rapid anthropogenic climate change. Understanding the adaptive response landscape of both plastic and selective processes would greatly improve the forecasting of the local and regional effects of global warming on plant species, particularly those with widespread ranges of distribution such Antarctic vascular plants.

It is important to mention that future climate change will not only raise temperature but will also increase liquid precipitation. For instance, *Turner et al. (2009)* documented increases in precipitation in the Antarctic Peninsula. It is important to acknowledge that in our experimental design plants we only evaluated the response of plants to warming,

maintaining irrigation similar for both treatments. As shown in *Torres-Díaz et al. (2016)* warming tend have lower effects than watering or combined warming plus watering on *C. quitensis* fitness. Therefore, our experimental design might have underestimated the effects of future climate change on *C. quitensis* net photosynthesis, growth and reproductive effort.

## Final remarks

It is important to note that our study is not free of limitations. Natural environmental conditions are almost impossible to reproduce in growth chambers, thus our estimations of plant performance could differ from those we would find in the field. Natural conditions were not fully mimicked in our experimental setup and future experimental research under field conditions could provide more accurate predictions of the effects of warming on *C. quitensis* performance. Field experiments using open top chambers (OTCs) would have been the more realistic way to study the future responses of *C. quitensis* to environmental warming. Nonetheless, as shown by *Torres-Díaz et al. (2016)* the responses of *C. quitensis* to future climate change can be complex, modulated by abiotic and biotic interactions, and additive and even antagonistic effects may affect the results of laboratory experiments.

## ACKNOWLEDGEMENTS

We thank INACH, the Chilean Navy and the Arctowski Polish Antarctic Station for their logistical support. We thank Manuel Balaguer for data sampling in the simulated global warming experiments. This article contributes to the SCAR biological research programs "Antarctic Thresholds—Ecosystem Resilience and Adaptation" (AnT-ERA) and "State of the Antarctic Ecosystem" (Ant-Eco).

### Funding

The Chilean Antarctic Institute (INACH) RT-11-13 and PII 20150126 projects financed this study. The funders had no role in study design, data collection and analysis, decision to publish, or preparation of the manuscript.

### Grant Disclosures

The following grant information was disclosed by the authors:
The Chilean Antarctic Institute: RT-11-13, PII 20150126.

### Competing Interests

The authors declare there are no competing interests.

### Author Contributions

- Ian S. Acuña-Rodríguez conceived and designed the experiments, performed the experiments, analyzed the data, contributed reagents/materials/analysis tools, wrote the paper, prepared figures and/or tables, reviewed drafts of the paper.

- Cristian Torres-Díaz contributed reagents/materials/analysis tools, wrote the paper, reviewed drafts of the paper.
- Rasme Hereme contributed reagents/materials/analysis tools, prepared figures and/or tables.
- Marco A. Molina-Montenegro conceived and designed the experiments, performed the experiments, analyzed the data, contributed reagents/materials/analysis tools, wrote the paper, reviewed drafts of the paper.

### Field Study Permissions

The following information was supplied relating to field study approvals (i.e., approving body and any reference numbers):

All plants were collected with permission from the Chilean Antarctic Institute (INACH).

### Data Availability

The raw data are presented as Supplementary Material.

### Supplemental Information

Supplemental information for this article can be found online at http://dx.doi.org/10.7717/peerj.3718#supplemental-information.

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
