# Peer review of "Asymmetric responses to simulated global warming by populations of Colobanthus quitensis along a latitudinal gradient"

_PeerJ, doi:10.7717/peerj.3718_

## Round 0.1 · original submission · Major Revisions

You need to answer the comments made by both referees regarding the design, both by giving a better description and by discussing the limitations with respect to precipitation and radiation.

I also ask you to present the data - as it is now, one only has the means/SEs, and no idea if there are outliers, how the variation is distributed, etc.

One referee also suggested to take into account initial plant size, this is easy to do using analyses of covariance.

I would also ask you to give results with more consistent number of significant digits (eg Table S2) or less digits. It is a bit meaningless to write 246.92 when the SE is 12 - the result is that the table is hard to read.

Reviewer 1 ·

Basic reporting

Abstract could be a little clearer when framing the justification, but general presentation and language are acceptable. Methods need much more detail and some of data is a little ambiguous. Further presentation of data, rather than just differnces, would be helpful.

Experimental design

Design seems reasonable, but difficult to fully assess without full methodological detail.

Validity of the findings

Difficult to fully assess without further detail about methods. Some inconsistency with argument is glossed over in Discussion. A little more equivocation could be added

Additional comments

The manuscript describes an investigation, which aims to predict the likely effects of global warming on populations of Colobanthus quitensis. The investigation is interesting, but is somewhat lacking in detail about methods and would benefit from some attention to minor inconsistencies in the Discussion.

More specifically:

L25 to 27 I’m not sure that I follow the reasoning here. Doesn’t the warming just mean that the physiological optimum temperature is moving south? The described warming experiments increase the temperature from current to future temperatures. Any increase of performance driven by this temperature increase suggests a likely expansion of local populations, but does not necessarily indicate a southerly expansion. A southerly expansion under global warming is, of course, indicated by the survival of the most southerly current populations under current conditions. I think that it would be helpful to the reader to clearly point out this distinction.

L120 to 127 This seems to be Introduction not Methods

L128 to 135 It would be more complete to add longitude as well as latitude for locations.

L137 to 144 Much more detail about all methods is required.

L154 to 176 Much more detail required. What was light intensity when not 20 micromoles? What light intensity was used for photosynthesis measurements? C. quitensis has a tricky morphology for measuring photosynthesis and leaf area. How exactly was this achieved? Exactly how was growth measured (taking into account soil moisture, plant water content etc.)? Were plants all exactly the same size at the start?

L198 to 199 It would be helpful to give actual mean photosynthesis for the different times and locations (maybe as a Supplementary table).

Fig. 2 are units m-2 leaf area?

L202 to 206 and Fig. 3 Is biomass fresh weight? The absolute growth increment (and flower number) is likely to be strongly affected by initial plant biomass. Were all plants the same size at the start? What is the response if relative units are used (% increase; flowers per unit biomass)?

L235 to 236 I think that it is at least worthy of mention that the most abundant pigment (by weight; it would also be helpful to use moles), did not have a clear relationship with latitude.
L248 to 250 Photosynthesis was not higher in the most southerly population.

·

Basic reporting

I have some comments on the experimental design.
Global change in the Antarctic Peninsula is not only global WARMING. In the publication Turner et al. Antarctic Climate Change and the Environment (2009) it is clearly shown that in the Peninsula precipitation has also changed significantly, this part of the Antarctic is getting wetter. This will no doubt having a significant effect on Colobanthus. This is not taken into account in the experiment. It should at least be mentioned in the text and the potential effects should be discussed. The experiment is done under a light regime of 20 mumol.m-2.s-1. That is only 1% of the irradiation the plants receive in nature. This very low light regime may cause artefacts and may influence the results.

Experimental design

The experimental design in itself is fine. I refer to my former comments, as I would have liked some alterations

Validity of the findings

The findings are well reported and discussed.

Additional comments

I have a number f small text errors:
line 24: Antarctic with a capital A
Line 53: delete polar regions including, there is only the Arctic and the Antarctic after all
line 62: delete et al. after Convey
line 73: Sexton et al.: is it 2012 or 2014 (as in the ref. list)
line 129 count??
line 181: R-Core Team 2015 should be put in the reference list
line 224: it is Demmig (without n) - Adams
line 234: I wonder if the increased precipitation produces a dryer environement
line 238: Cavieres et al. is it 2016 or 2015 as in the reference list
line 251: Kelly & Goulden 2008 is not in the reference list

---

## Round 0.2 · Minor Revisions

I am happy with most of the changes that you made but you need to make the following changes - please check the language carefully before you resubmit the paper:

- l. 23: facilitating instead of propitiating
- l. 36. "Overall, we found a clinal trend ..."
- l. 32: the and not their
- l. 33: these and not theses
- l. 36: remove -by warming-
- l. 40: remove ", mainly in those from southern"
- l. 41: remove "those", and replace "could be improved in" by "could increase"
- l. 60: antarctic pearlwort, not pearlworth.
- l. 101: studies and not designs
- l. 196: remove "independent".
- you kept two many significant digits in Table 1 (i mentioned table S2 in my previous comments but they apply to all tables...). Again writing 358.28 when the SE is 24 is not very useful and make the table hard to read. Simplify it.

·

Basic reporting

The paper is well revised

Experimental design

Sufficiently revised

Validity of the findings

the findings are correctly reported

Additional comments

The authors have sufficiently addressed the remarks made bij the reviewers.
This manuscript can be accepted and published.

---

## Round 0.3 · accepted · Accept

Thanks for making the required changes in this second version.